# Utilizing Smartphones for Approachable IoT Education in K-12 [note 1]

**DOI:** 10.3390/s22249778

**Published:** 2022-12-13

**Authors:** Devin Jean, Brian Broll, Gordon Stein, Ákos Lédeczi

**Affiliations:** Computer Science Department, Vanderbilt University, Nashville, TN 37235, USA

**Keywords:** IoT, mobile devices, sensors, user interaction, block-based programming

## Abstract

Distributed computing, computer networking, and the Internet of Things (IoT) are all around us, yet only computer science and engineering majors learn the technologies that enable our modern lives. This paper introduces PhoneIoT, a mobile app that makes it possible to teach some of the basic concepts of distributed computation and networked sensing to novices. PhoneIoT turns mobile phones and tablets into IoT devices and makes it possible to create highly engaging projects through NetsBlox, an open-source block-based programming environment focused on teaching distributed computing at the high school level. PhoneIoT lets NetsBlox programs—running in the browser on the student’s computer—access available sensors. Since phones have touchscreens, PhoneIoT also allows building a Graphical User Interface (GUI) remotely from NetsBlox, which can be set to trigger custom code written by the student via NetsBlox’s message system. This approach enables students to create quite advanced distributed projects, such as turning their phone into a game controller or tracking their exercise on top of an interactive Google Maps background with just a few blocks of code.

## 1. Introduction

Most of the applications we use on our computers and mobile devices every day are distributed and use the Internet to provide their functionality. Networked sensors and actuators—the Internet of Things (IoT)—are also becoming ubiquitous, with smart homes and health monitoring leading the way. However, hardly any of the enabling technologies are taught in introductory computer science (CS) classes in K-12. There do exist classes and makerspaces where some students are exposed to embedded computers, providing opportunities to program Raspberry Pis or micro:bits with simple sensors and actuators, such as LEDs, using connectivity based on either a USB cable or Bluetooth. However, while these experiences are fun, they are fairly disconnected from the IoT that otherwise surrounds us. In addition, not many schools offer these types of classes due to cost, logistics, and a lack of teachers who have experience with these tools.

However, over 84% of teenagers in the United States already own a mobile device [1] that comes with a rich set of powerful sensors, including an accelerometer, gyroscope, microphone, camera, GPS, and many more, and is Internet-enabled out of the box. Thus, smartphones offer an excellent opportunity to expose students to key networked sensing topics, such as polling and streaming access paradigms and event-based computing, and make computing more engaging by enabling students to be creators and not just users of compelling applications. However, one important question remains: how can we make these powerful technologies accessible to novice programmers?

This work introduces PhoneIoT, an open-source mobile app for Android and iOS that allows users to programmatically access their own smartphones and tablets as IoT devices through NetsBlox [2], a block-based programming environment based on Snap! [3]. NetsBlox introduced two powerful networking and distributed computing abstractions to block-based languages: Remote Procedure Calls (RPCs) and message passing. RPCs make a rich set of online services and data sources accessible to student programs, such as Google Maps, earthquake data from USGS (United States Geological Survey), climate change datasets from NOAA (National Oceanic and Atmospheric Administration), The Movie Database, gnuplot (accessed online through NetsBlox so that students need not have it installed), and many more [4]. Message passing lets NetsBlox projects running anywhere in the world communicate with one another through the exchange of custom packets of structured data, making it possible to create multi-player games and other distributed programs. RPCs and messages are also used to control WiFi-enabled devices such as educational robot vehicles [5].

PhoneIoT provides two main features: the ability to read and/or stream live sensor data, and the ability to create an interactive, configurable user interface on a mobile device. Together, these features address the fundamental requirements of an educational IoT tool, including the ability to access sensors through common paradigms such as polling and streaming, as well as a way to directly interact with the device to create more engaging projects, such as integrating sensor data and custom user input to provide feedback on the device’s display. Importantly, all PhoneIoT features are accessible through the typical NetsBlox primitives of RPCs and message passing, making its usage both simple for novices and familiar for existing NetsBlox users. By using a simple yet powerful block-based interface, it becomes possible to abstract away much of the complexity of networking and distributed computing while allowing students to explore and learn the most important concepts in a convenient framework.

The primary contributions of this paper are: an overview of user-aware PhoneIoT design choices that were made to facilitate targeted curriculum topics and ease of use, as well as a preliminary, proof-of-concept study with students who used PhoneIoT in 1–2 week IoT coding camps. We note that this is an extended work from the 2021 conference paper Your Phone as a Sensor: Making IoT Accessible for Novice Programmers [6].

## 2. Previous Work

There are several existing approaches that allow for the creation of standalone mobile apps that can be constructed online with a block-based programming interface, including Thunkable, App Inventor, and Kodular (formerly known as AppyBuilder) [7,8,9]. Pocket Code, part of the Catrobat project, is similar to these, although its app designer is built into the app itself and is more focused on creating games and simulations [10]. Thunkable is perhaps the most similar to PhoneIoT, as it allows access to Internet resources (e.g., cloud-based speech recognition and translation utilities) similar to NetsBlox services, as well as several onboard device sensors, such as the accelerometer and gyroscope.

However, PhoneIoT is fundamentally different from these projects in that it does not aim to be an app creation tool; rather, the custom controls in PhoneIoT are merely a means of interacting with NetsBlox code running in the browser on the student’s computer. That is, Thunkable and similar projects are not tools for teaching distributed computing or IoT, as all user interaction and sensor data are kept internal to the device running the app. Additionally, because PhoneIoT is tailored to a distributed computing environment, it offers more possibilities for creating engaging educational projects. For instance, PhoneIoT could be used to turn a phone into a custom game controller, with accelerometer input and soft (virtual) buttons on the phone’s screen making sprites move or shoot on the computer’s screen. The phone could also be used to control real robots [5] in the same way, using a single NetsBlox program to control multiple components of a distributed system: one or more mobile devices, one or more robots, and the laptop running the project code, creating an engaging distributed application.

Another project similar to PhoneIoT in terms of intent and network architecture is Sensor Fusion, an education-focused project which collects sensor data from a mobile device and streams it to a computer for analysis [11]. This is similar to the core sensor-based functionality of PhoneIoT but is more heavily focused on a scientific perspective, namely sensor fusion, which is the combination of data from multiple sensors to achieve greater accuracy or precision. In contrast, PhoneIoT is part of a distributed computing environment, empowering students to utilize incoming live data streams, as well as content from other NetsBlox services, and reconfigure the phone’s display in real time based on the desired application (e.g., a game controller, data viewer, or fitness tracker). This is not possible with Sensor Fusion, as its display and interactive components are not configurable. PhoneIoT’s programmability is a key factor in creating engaging educational projects for young learners.

## 3. PhoneIoT

Mobile devices already come with a wide variety of hardware sensors, from simple cameras/microphones, accelerometers, and gyroscopes to more specialized hardware such as contact pressure sensors or relative humidity detectors. Although a typical device does not contain all of these potential sensors, there are several sensors that are reliably present even on older devices, simply due to basic system requirements. These include an accelerometer (which is typically used for automatic landscape/portrait screen rotation) and, for smartphones, a microphone and proximity sensor (to disable the touchscreen when the phone is held to the user’s ear). While not essential for core device functionality, virtually all modern smartphones and tablets also have a camera, although access to this sensor through PhoneIoT is handled differently due to privacy concerns (see Section 3.1). Additionally, through services such as Google’s Fused Location Provider API, any mobile device connected to the Internet can retrieve live location data, if not by GPS, then by estimation from the connected local network.

The PhoneIoT app is capable of accessing all of these common sensors and more. If a sensor is not present on the device, is disabled, or otherwise blocked by app permissions, it is simply logically disabled as a target for IoT interactions through the NetsBlox interface. PhoneIoT continuously monitors data from all available sensors and makes it available to the NetsBlox server when requested by an authenticated student’s program. The server also handles other specialized requests, such as GUI configuration and forwarding user interactions as messages to linked NetsBlox clients. Figure 1 visualizes this system architecture.

### 3.1. Privacy

Due to its extensive access to live device sensors, PhoneIoT introduces a number of potential privacy issues, primarily due to exposing access to live location data, the camera, and the microphone over the Internet. These concerns become especially important given that minors will be using the app, as it is largely meant to be a K-12 educational tool. For instance, it would not be acceptable for someone to be unknowingly tracked or spied upon through the NetsBlox interface due to forgetting to close the app. Because of this, unless explicitly requested with the “run in background” setting in the menu, the app ceases communication with the server and rejects all incoming requests upon being put into the background (minimized) or when the device’s display turns off, e.g., due to inactivity. As a further precaution, the app generates a random password which must be provided for any IoT interaction. For added security, this password is set to expire one day after generation (or upon user request), at which point a new random password is created, effectively cutting off any active connections. This one day window is sufficient for most uses of the app while still providing necessary privacy guarantees.

The password and expiry behavior is sufficient to make location data reasonably secure, but the camera and microphone could still be problematic. To solve the microphone issue, PhoneIoT only exposes the current volume level, rather than the actual waveform/content. To solve the camera issue, only images stored in image displays (described in Section 3.3) are accessible; that is, the app does not allow a network request to take a new picture from the camera without user interaction. We believe these behaviors are sufficient to allow any K-12 audience to use the app while still affording them reasonable internet privacy.

### 3.2. Network Exchanges

As the app was meant to be used by young audiences in classroom settings, connecting PhoneIoT to a NetsBlox server deployment is very easy, only requiring a single button press from the app menu, which is already open when the app is started. Once pressed, the app connects to the server, announces its presence as an IoT device, provides the server with a unique identifier for further communications, and begins accepting network requests forwarded from the server.

The targeted server address is displayed as a URL in the app menu; this defaults to the primary NetsBlox server deployment but can be configured to any address. This is especially important in some classrooms around the world where a stable high-speed internet connection is not available, in which case a local deployment of NetsBlox can be used on the local network. The server address field is persistent across app restarts, so classrooms in these less-ideal circumstances would only have to configure their app settings once.

The UDP protocol was selected for PhoneIoT’s server interactions both for speed and because PhoneIoT’s data exchange model is already packet-based, making UDP a more natural model than streaming protocols such as TCP. Although UDP has the potential issue of dropping packets, for our purposes, this is actually desirable due to providing real-world lessons on error-handling in fallible network transactions. For instance, an early project for students could be to make robust wrappers for some PhoneIoT functions by repeating the operation until it succeeds.

The networking primitives used by the NetsBlox side of PhoneIoT are composed of “messages”, the same concept used throughout NetsBlox. In essence, a “message” is a structured block of data that is identified by name and has a set of fields associated with it. Messages can be sent with the “send msg” block and received (typically on a different computer) with a “when I receive” block. As an example, there is a default message type called “message” which has a single field called “msg”. Figure 2 shows a simple example of how to send and receive a message of this type.

PhoneIoT provides two primary ways of accessing sensor data: polling for instantaneous values through explicit RPC requests, or streaming up to date values by registering a message type as a sensor update event that the device will send periodically based on the requested update interval. The explicit request style is similar to other pre-existing networking APIs in NetsBlox, and thus is a good introductory point for using PhoneIoT. However, in practice, many real-world IoT devices are accessed by continuous data streams, so lesson plans involving PhoneIoT quickly transition to this method.

Because PhoneIoT has the chance of dropped packets due to using the UDP protocol, performing polling in a loop can result in an error every once in a great while, depending on the network connection; this would have to be checked by students to avoid bugs in their code. Thus, streaming access can be introduced naturally as a more elegant solution, removing the need for both the loop and error checking code, as well as halving latency by inverting the problem and instructing the PhoneIoT device to send periodic update messages to the student’s project, rather than the project having to request each one. In this way, dropped packets go from causing errors to being simply absent update messages which are simply ignored by the student’s project without issue. Figure 3 shows example code which registers for and receives sensor updates from the accelerometer every 100 ms.

From examining Figure 3, one detail that may not be clear is how to know the name of the sensor to listen to, in this case “accelerometer”. For the most part, the names of the various sensors are identical to their polling RPCs; for instance, the “getLinearAcceleration” RPC has a matching sensor name of “linearAcceleration”. However, this is not always the case, in particular for soft sensors that PhoneIoT adds on top of the existing hardware sensors; for example, the “getCompassHeading” RPC’s sensor is in fact “orientation” (matching the “getOrientation” RPC). This information is available to users through the RPC help menu, which can be accessed by right clicking on a “call” block configured to, e.g., a polling RPC and selecting “help”. Figure 4 shows the help information for the “getCompassHeading” (polling) RPC, which displays the name of the sensor/message type for the streaming access method, as well as the available fields that can be received on each update.

### 3.3. Custom GUI Controls

A key feature of PhoneIoT is its customizable interactive display. The static GUI for the main screen of the PhoneIoT app is intentionally minimalistic, containing only a button to toggle the pull-out app menu that contains all other static controls. Importantly, the menu is where the device’s id and password are shown, as well as the controls for connecting to the NetsBlox server; this menu is shown in Figure 5a. When the menu is closed, the entirety of the screen, aside from the menu toggle button, is a single blank canvas which can be populated with content via various RPCs from the user’s program.

PhoneIoT supports many standard GUI control types, such as labels, buttons, text fields, image displays, and toggle switches, as well as some controls tailored for designing game controllers, such as virtual joysticks and touchpads. Each of these controls has fully customizable text content, location, size, color, orientation, and several other options depending on the specific control. A non-exhaustive example of custom control types is given in Figure 5b,c.

An important consideration when designing PhoneIoT was to ensure that projects could be easily shared between students, who are potentially using different devices. Unfortunately, all of these different devices have different screen resolutions. To counteract this, PhoneIoT uses a relative, percent-based scale for specifying the *x*/*y* position and the width/height of controls. Specifically, PhoneIoT uses the standard GUI coordinate layout where (0,0) is the top left corner of the canvas and (100,100) is the bottom right. Additionally, the app automatically scales fonts depending on the DPI of the display, which allows fonts to be approximately the same size on all screens. These simple accommodations result in a coordinate system that is easy for students to use and is roughly invariant of the specific device display being used (up to mostly-minor aspect ratio stretching).

Being able to display custom content on the device is all well and good, but many of these types of controls are intended to facilitate receiving input from the user, such as button presses and joystick movement. To facilitate this, the various RPCs that are used to add controls to the device accept an optional configuration setting called “event”, which is the name of a message type that the device will send when a user interacts with the control. For instance, button events are triggered when pressed, text fields are triggered when the text is modified and submitted (i.e., there is not a separate update for each keystroke), image displays are triggered when a new image is saved in them from the camera (which can only be done manually by the user—see Section 3.1), and joysticks/touchpads send an event when initially touched, continuously while moving (currently throttled to 10 Hz), and when released. Figure 6 gives an example of creating a joystick control with an event called “joyMoved” that is triggered each time the stick is moved and displays the *x*/*y* position of the stick on the NetsBlox stage.

Message names for GUI events are customizable, but each type of control sends a different pre-determined set of values that users can receive by including them as fields on the message type. For instance, text field events include the new text, joystick/touchpad events include the *x*/*y* position, as well as a “tag” field that is one of “down”, “move”, or “up” to determine the type of interaction, and so on. Additionally, all events send the ID of the control (which can also be obtained by the return value of the RPC used to create the control) and the device ID on which the control was located (to differentiate controls on projects that configure multiple phones with the same GUI layout, such as a quiz game or group chat app). The specific information concerning what fields are available for each type of control can be found in the detailed documentation that is linked at the bottom of the (basic) RPC help menu. An example of this full, detailed documentation for the “addJoystick” RPC is shown in Figure A1.

When a control is created, it is automatically assigned a control ID, which is returned by the RPC that was used to add the control (but if not needed, can be discarded by using a “run” block rather than a rounded “call” block). This control ID can then be used after creation to get and set state information about the specific control, or to delete the control while leaving all others intact. For instance, RPCs that contain text (e.g., labels, text fields, and buttons) can be used with the “getText” and “setText” RPCs, image displays can be used with “getImage” and “setImage”, toggle-based controls can be used with the “getToggleState” and “setToggleState” RPCs. This type of dynamic update and query behavior after control creation is vital, and can be used to perform tasks such as real-time updating of information displayed on the device screen.

This interactive component is important for teaching IoT to younger K-12 audiences because it immediately gives the students a useful tool related to things they already know, such as game controllers or content sharing with text/image displays. Due to how important phones are to today’s youth, introducing them to new ways of engaging with and controlling their devices can be especially motivating for continued interest in CS topics. The networking and IoT components are added to this to provide even more functionality and to teach the concepts to an already eager audience as a “side effect”.

## 4. Example Projects

This section will cover several example projects to demonstrate how phone-based IoT through the NetsBlox platform enables powerful applications with very little code or specialized knowledge.

### 4.1. GPS Tracker

The NetsBlox platform already supports many online services, one of which is Google Maps. With this service, a program can obtain and display a map of the current location, specified by latitude and longitude, or obtain the screen position of a latitude and longitude point on the map and vice versa. By reading live GPS data from a mobile device running PhoneIoT, it is possible to track the location of the device on a map and use NetsBlox’s built-in drawing utilities (inherited from Snap!) to plot the course. Thus far, this has all been performed on the NetsBlox client (for user logic and drawing) and the NetsBlox server (for performing API requests), with the device running PhoneIoT only being used as a sensor. However, by using PhoneIoT’s custom GUI elements, we can add an image display to the screen and send periodic updates to the mobile device. Essentially, this creates a stripped-down form of the Google Maps front-end that can be built in under ten blocks. See Figure 7 for the blocks that set up the display, Figure 8 for the update logic running in NetsBlox, and Figure 5c for the custom GUI shown on the mobile app.

### 4.2. Accelerometer Plotter

A common topic in introductory IoT is analyzing a live data stream coming from a device. We have already seen that receiving a data stream from PhoneIoT is as simple as one RPC call, followed by listening for NetsBlox messages. Once the data are received, students can perform whatever analysis is needed and output results to their NetsBlox client display. A simple project that could be conducted on a student’s first day of working with PhoneIoT is to receive live accelerometer data and plot its *x*, *y*, and *z* components. This can be done with the Chart service, a pre-existing NetsBlox service for generating graphs from data points. Figure 9 shows the code required to do this, as well as the NetsBlox display (running on the student’s computer) after the phone was picked up from rest, rotated slowly around one axis, then another, and finally dropped onto a pillow.

### 4.3. Robot Remote Controller

One of the many services supported by NetsBlox is RoboScape, which allows students to interact with physical robots that can be shared in a classroom setting. Students can send commands to the robots via RPCs, and can asynchronously receive messages from robots based on basic sensor input or other events. The interfaces for these features are very similar to PhoneIoT (in fact, PhoneIoT copied this interaction pattern due to its proven success with students [5]), with commands being issued by normal RPCs that are proxied through the NetsBlox server and event-based messages being sent back from the device asynchronously after registering to receive them with the “listen” RPC. A later development was RoboScape Online [12], which is a fully virtual environment embedded in NetsBlox as an extension; it gives access to virtual robots that connect into the same existing RoboScape networking layer and so can be controlled by the very same student programs that would be used with their physical counterparts. RoboScape Online makes it possible to give students access to any number of robots (i.e., classroom sharing is no longer required), works well in virtual classroom settings, and offers unique motivating features such as automated scoring for challenges and access to virtual simulations of many different types and combinations of sensors (e.g., LiDAR, radiation sensor, light sensor, GPS, or compass). See Figure 10 for an example of the RoboScape Online environment.

Both PhoneIoT and RoboScape Online were included in the IoT version of two 1-week Computer Science Frontiers (CSF) camps. On the first day of the camps, students were introduced to NetsBlox basics, such as control flow, lists, loops, RPCs, and message passing. The following two days introduced RoboScape Online and had the students complete several increasingly complex robotics challenges, with a focus on autonomous control based on various virtual sensors. The following day introduced PhoneIoT, which was made easier by students already having become familiar with the same interaction schemes through RoboScape and RoboScape Online. Finally, the week concluded with two projects which had students turn their phones into custom robot remote controllers by combining RoboScape with PhoneIoT’s interactive GUI and sensory features. Here, we present one such controller that students could create.

The concept for this controller is to have a “throttle” slider on the phone screen that can be used to control the robot’s speed in real time; meanwhile, orienting the robot in a particular direction will be done via the phone’s internal compass using the orientation sensor. The only RoboScape RPC we will need is “send”, which allows us to issue simple text commands to the robot; e.g., we can control movement by sending the command “set speed L R” where “L” and “R” are the speeds for the left and right wheels, respectively, which are integers between –128 and 128. To match the robot’s virtual heading to the phone’s real heading, the only additional piece of information required is querying the robot’s heading. This can be done with the PositionSensor service’s “getHeading” RPC; all RoboScape Online sensors that were not present in RoboScape proper are hosted as dynamically-created “Community” services such as PositionSensor. Figure 11 contains all the code needed to create the throttle slider, listen to updates from the phone’s orientation sensor, continuously turn the robot to face the desired direction, and proceed forward to the target with the desired speed; also included is an image of the configured GUI on the phone screen. Note that, although this would be a complex robotics task to perform from scratch, PhoneIoT and RoboScape Online provide sufficient abstractions that students can create these advanced behaviors with only a handful of blocks.

## 5. Use with Students

PhoneIoT is still relatively new and has not had thorough analysis with students. However, we were able to include some of its curriculum in two 1-week CSF camps (as described in Section 4.3), as well as the second week of a 2-week fully-remote cybersecurity camp involving 17 high school students with no required previous programming experience. The first week of the camp consisted of an introduction to NetsBlox basics such as RPCs and message passing. This was primarily used to control 3D simulated robots in RoboScape Online and perform various manual and autonomous robotics challenges. During week two, students were introduced to PhoneIoT through several projects, including an “Avoid the Holes” game, which used sensor streaming with the accelerometer to apply a tilting-based force to a ball sprite rolling around through a maze of holes. Students were later introduced to the interactive graphical display by implementing the GPS Tracker project shown in Section 4.1. The remaining three days of camp consisted of fusing PhoneIoT and RoboScape by converting their phones into custom robot controllers, an example of which was provided in Section 4.3. We note that the specific robot controller example given in Section 4.3 was not a student project; however, its complexity is comparable to other robotics tasks that students completed. All students successfully implemented the projects involving PhoneIoT; additionally, students seemed to pick up PhoneIoT relatively quickly, which could be explained by the fact that PhoneIoT’s NetsBlox interfaces are very similar to RoboScape.

The camp included pre- and post-tests; however, the questions were primarily gauging student interest in various aspects of CS, and none of the questions were specifically about PhoneIoT. The results of the study included increased scores for student interest in computer networking, data analysis for scientific issues, desire to use CS in their careers, confidence in CS, desire to pursue CS in college, and several other similar categories. However, the sample size was small (17) and response rates for the post-test were even lower (10). Coupled with the low sample size and the lack of questions specifically targeting PhoneIoT, it is unclear from this study what effect PhoneIoT alone had on the results. However, in a few free response portions, several students expressed that they enjoyed PhoneIoT, with over half (6) including PhoneIoT projects in their top three most useful projects of the camp. Because of these reasons, in the future, we intend to pursue another study with a new pre-/post-test including questions specifically concerning PhoneIoT to better gauge its effectiveness with students.

### CSF Course

The CSF camps previously mentioned have been 1- and 2-week studies; these are effectively contracted versions of existing CSF curriculum, which is quite extensive in its entirety. In fact, it has recently received approval to become a full-fledged high school course, and is being piloted in Nashville, Tennessee’s Martin Luther King Jr. Magnet High School. The course consists of four 9-week modules: Distributed Computing, IoT, and Cybersecurity (including PhoneIoT and RoboScape Online, among other topics), Artificial Intelligence and Machine Learning, and Software Engineering. The full range of curriculum is beyond the scope of this paper, but we will overview the IoT and Cybersecurity module, which is largely hands-on, with students solving challenges or creating original applications using what they have learned in previous lessons/projects.

In the first week of the module, students are introduced to the fundamental concepts of IoT through the use of several tools such as ThingSpeak, which provides unified access to many different types of IoT sensors around the world [13]. The following two weeks introduce PhoneIoT, with the first week consisting of sensor-based projects, and the second week being GUI-focused. Some of the PhoneIoT projects that are covered are similar to those seen in Section 4, though they are typically more in-depth versions with additional features and/or creative extensions added by students as part of open-ended programming projects. The next two weeks focus on RoboScape Online with various manual and autonomous tasks, including its fusion with PhoneIoT for creating custom remote controllers (see Section 4.3). The next two weeks cover several cybersecurity topics with RoboScape, which are beyond the scope of this paper. In addition, finally, the last 1–2 weeks of the module (depending on pacing) are reserved for extended individual or team projects; these are meant to be large creative projects that culminate all the content learned up to that point and give students more time and creative liberty than the other, more focused open-ended tasks throughout the module.

## 6. Conclusions

In this brief overview, we have shown that PhoneIoT is a low-cost method for allowing K-12 students to access device sensors for learning the key concepts of networked sensing. These concepts include API requests in fallible conditions, methods for handling failures, sensor data processing, event based programming via message passing, and potentially many other topics depending on usage (e.g., error mitigation for noisy sensor modalities such as GPS location). Additionally, the custom display on the phone allows students to come up with novel ways to interact with their code running on NetsBlox. We believe students will find PhoneIoT an enjoyable educational tool that will allow them to envision and create innovative distributed applications. Along the way, they will learn important cutting edge computing concepts rarely taught in K-12 today. 

## Figures and Tables

**Figure 1 sensors-22-09778-f001:**
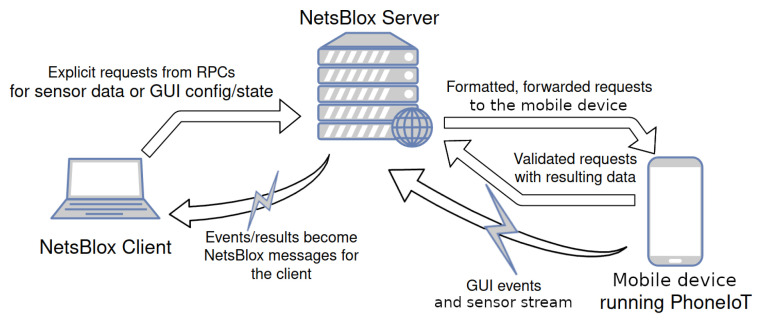
A visualization of backend network interactions between a NetsBlox client, the NetsBlox server, and a device running PhoneIoT.

**Figure 2 sensors-22-09778-f002:**
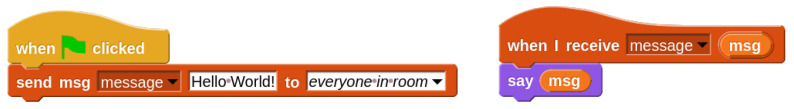
Example of sending and receiving messages in NetsBlox.

**Figure 3 sensors-22-09778-f003:**
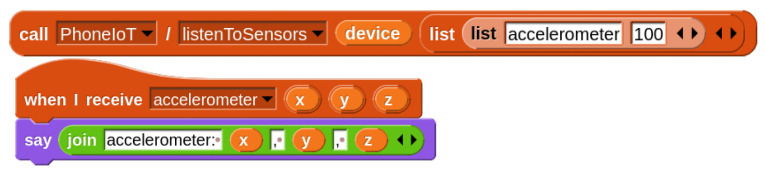
Registering for and receiving accelerometer updates at 10 Hz.

**Figure 4 sensors-22-09778-f004:**
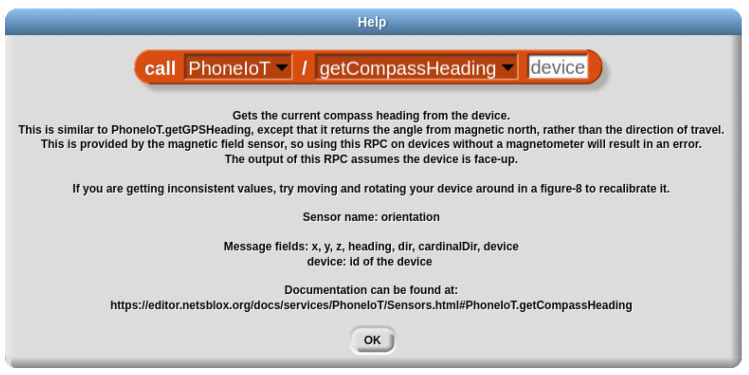
Example help documentation for the “getCompassHeading” polling RPC. This includes the name of the sensor and message type details needed for streaming access.

**Figure 5 sensors-22-09778-f005:**
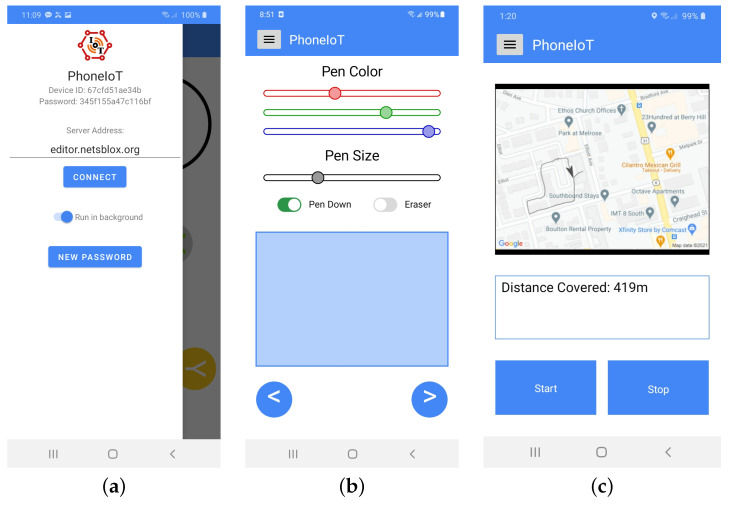
The PhoneIoT app menu (**a**) and two PhoneIoT apps with various controls (**b**,**c**).

**Figure 6 sensors-22-09778-f006:**
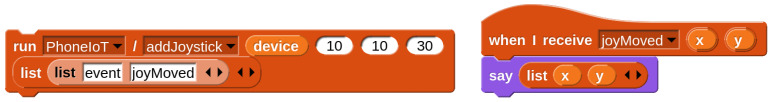
Example of creating a joystick control with an event called “joyMoved”.

**Figure 7 sensors-22-09778-f007:**
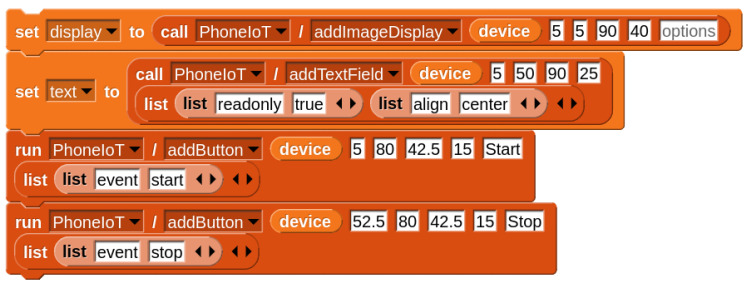
Blocks used to set up the PhoneIoT device display for the GPS tracker.

**Figure 8 sensors-22-09778-f008:**
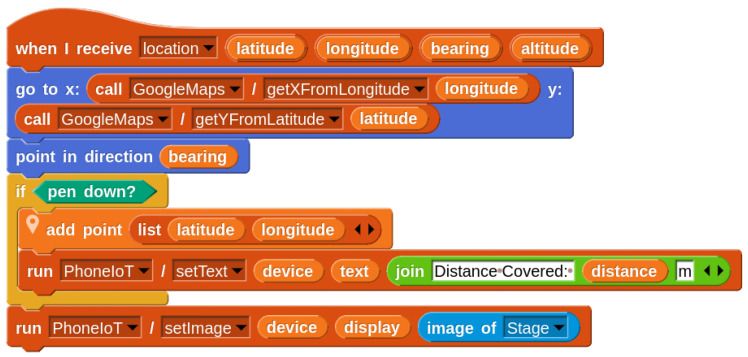
Location message handler. It reads live GPS data from the mobile device, plots the track on a map on the stage, and sends the map/track back as an image to the device. The “add point” custom block (function) consumes the location sequence and updates the total “distance” variable using a Google Maps RPC to get the distance between the current and previous locations.

**Figure 9 sensors-22-09778-f009:**
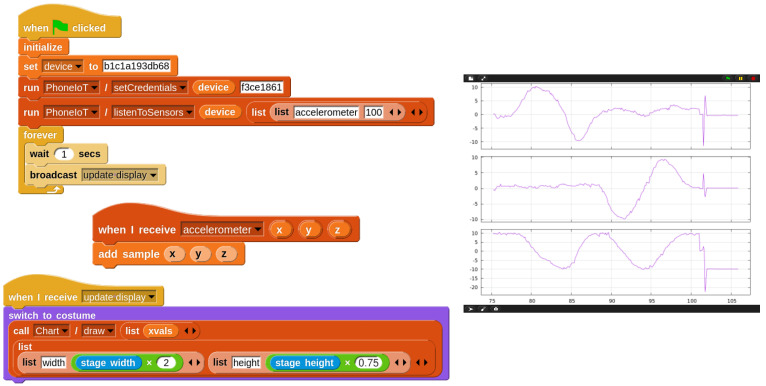
Code running the accelerometer plotter project. The “add sample” custom block (function) constructs three lists (xvals, yvals, zvals). The “update display” script is only shown for the *x* sprite.

**Figure 10 sensors-22-09778-f010:**
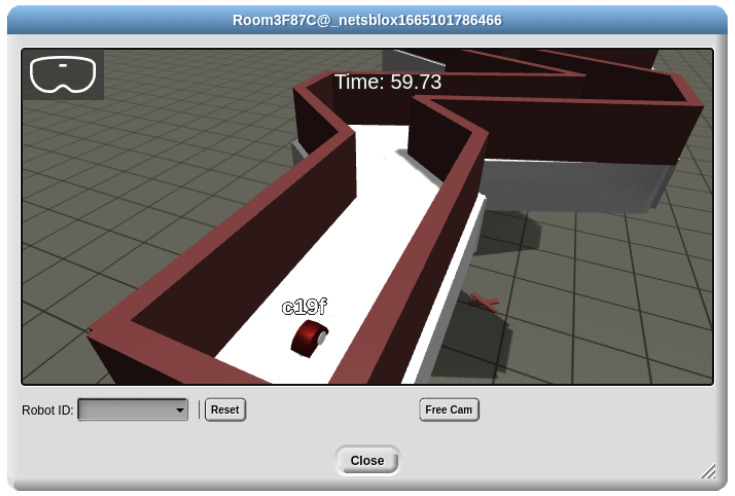
RoboScape Online with an environment that has students race through a bending hallway.

**Figure 11 sensors-22-09778-f011:**
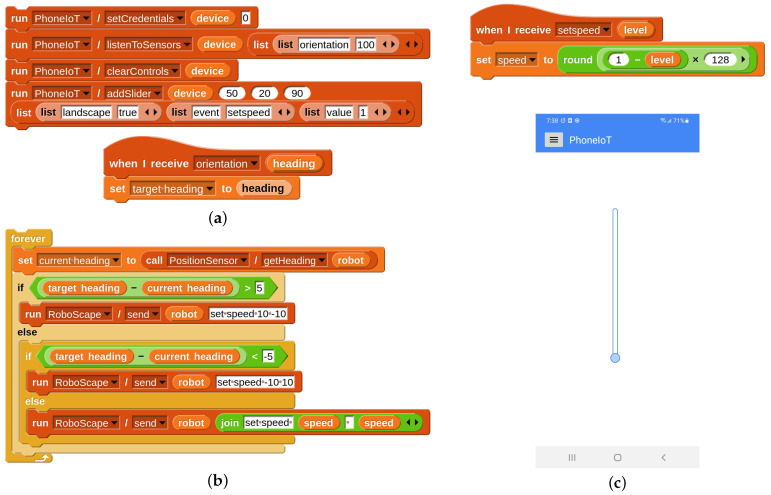
An example project that acts as a custom robot remote controller using PhoneIoT. (**a**) PhoneIoT code to set up the speed slider and start receiving target headings from the phone’s orientation/compass sensor; (**b**) RoboScape code to match a target heading and advance forward when facing the correct way; (**c**) the PhoneIoT app with its configured interface for throttle control.

## Data Availability

Not applicable.

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
