# Peer review of "Utilizing Smartphones for Approachable IoT Education in K-12 [Author-notes fn1-sensors-22-09778]"

_sensors, 2022, doi:10.3390/s22249778_

Round 1
Reviewer 1 Report
Comment 1:
The paper appears like a project report. The paper needs to discuss the problem or question and challenges. The literature review needs to be connected to the challenge, drawing the knowledge gap or technology gap.
Comment 2:
In elsewhere, the problem of privacy, security (I.E., protection of practice), and resilience (ie, recovery of the damaged protection) in the mobile computing environment is discussed, see “Privacy, security and resilience in mobile healthcare applications” (enterprise information systems). The authors need to compare their work with this work at the concept level, though the present work looks like an implementation.
Comment 3:
The paper needs to explicitly discusses the contribution.
Author Response
- We have added a few sentences to the introduction section to mention that the main challenge is making basic IoT concepts like polling and streaming approachable for novice programmers, with some following sections describing features (e.g., sensors and gui) that have been brought down to the level of student use.
- In this paper, the security section is meant more as an end-user level of privacy/security detail, rather than rigorous coverage of a new approach to data security in general. We believe this confusion was caused by this paper being recommended for a different special issue topic than expected, which we are currently working with the editors to correct and move to a more relevant special issue.
- The core contribution is the development of the PhoneIoT tool and its set of features for educational applications, which we believe is now better covered by our additions to the introduction.
Reviewer 2 Report
This manuscript presents a PhoneIoT, i.e., a mobile app that makes it possible to teach some of the basic concepts of distributed computation and networked sensing to novices. I see the idea is good and sounded; however, many issues should be addressed according to the following comments:
1) The overall presentation, readability, and more results are mandatory. Please, correct the language problems, it is weak from the Grammarly and sequences of events, I catch 19 errors by using a personal program, and the authors should cure them carefully.
2) The "Abstract" section should be more intensively focused on the main idea directly and must contain the contribution of this manuscript supported with numerical result indicators. Further, the abbreviations of IoT and GUI should be defined first. Also, the author must avoid the use of the pronouns "we & our" in the whole manuscript.
3) The "Introduction" section should be enriched with up-to-date references, 13 references only are too weak, please add and cite the latest trends in the area of smart applications using IoT. E.g., Reliable industry 4.0 based on machine learning and IoT for analyzing, monitoring, and securing smart meters & Reliable Deep Learning and IoT-Based Monitoring System for Secure Computer Numerical Control Machines Against Cyber-Attacks & Effective IoT-based deep learning platform for online fault diagnosis of power transformers.
4) It is mandatory to check carefully all the abbreviation definitions, symbols, and standard units in the whole manuscript. I catch some errors and the other symbols are not defined.
5) Please, the authors should add various equations or a flowchart that describe the issue procedures.
6) The resolution and quality of result figures (e.g., Fig. 9) should be presented as close to the camera-ready format. Also, please don't use the symbol abbreviations on X-Y-axes, they must have the full name with their SI units.
7) The conclusion section should be more concentrated and supported by the numerical results. Also, the authors may propose some interesting problems as future work in the conclusion.
Author Response
- Unfortunately, as an education tool with limited access to students, we do not currently have more results than those that have presented. However, in the absence of additional results, the paper is intended to cover important features and design choices, with our existing results with students as a demonstration of feasibility for practical use in classroom settings. We have corrected some language in the Introduction section.
- We have updated the abstract, but do not currently have numeric results to include. We believe one issue is that this paper was suggested to a different special topics section than we expected, which we are working with the journal editors to correct and move to a more appropriate education-related special issues topic.
- We believe this issue of references is caused by the aforementioned issue of special issue topic. The more technical suggested references of deep learning and cybersecurity are beyond the purview of our intended education-related special issue. We are currently working with the journal editors to correct this.
- The plots shown in Fig. 9 are the direct output of a simple program that students would write. Adding additional information like units and axis labels would make the student program code more complicated, so it was omitted as this is not meant to be performance data about the app itself (just a demonstrated student project).
Reviewer 3 Report
This paper proposed Utilizing Smartphones for Approachable IoT Education in K-12, which aims to strengthen the IoT data processing and improve the usage of smartphones in K-12 system.
Overall paper is written in well and organized manner. However, I suggest some minor revisions for the authors.
• In Abstract, it would be better to indicate the quantitative percentages of the obtained results.
• At the end of the introduction section. Although authors have provided the main contributions but it would be better to explain a bit more in a form of paragraph.
• I have seen some irrelevant related work in Section 2. I suggest to revisit it and improve it.
• Experiments and their discussion should be presented in separate sub sections.
• How this research will be carried out by the future researchers, include some clear future directions in the conclusion section.
. Ref. are too few & need to include more latest state of art.
. Analysis section is also not up to the journal mark; plz enhance it.
Author Response
- Unfortunately, as an education tool with limited access to students, we do not currently have rigorous methods or numeric results to include. However, in the absence of additional results, the paper is intended to cover important features and design choices, with our existing results with students as a demonstration of feasibility for practical use in classroom settings.
- We have modified the introduction to better directly address the problems and contributions.
- One of the issues with references (e.g., not including state of the art papers) we believe is caused by the recommendation of this paper in a different special topic than we had expected. This paper is intended to be published in an education-related area, making the state of the art papers beyond our purview. We are currently working with the journal editors to correct this and move to a more appropriate special issue.
Round 2
Reviewer 1 Report
The revision is fine with me.
Author Response
Thank you for your review.
Reviewer 2 Report
Most of my concerns didn't answer. I encourage the authors to change the type of this manuscript from "Article" to "communication".
Author Response
We are still working with the editors to change this to a more relevant special issue topic, which addresses the lack of numeric results.
Reviewer 3 Report
References are too few & latest state of art is still missing in introduction, background & Analysis. Please enhance it.
Include your main contributions after section 2.
How this approach is better than others, I could not find enough justification to publish in high quality journals like sensors.
Author Response
We are still working with the editors to change this to a more relevant special issue topic. We do not include discussion of state of the art sensing methods because this is meant to describe the specifically education-facing design decisions for the tool. However, we do include an overview of several popular related apps, which could be considered state of the art in the education domain. Some of the benefits of PhoneIoT over existing approaches is explained in the background section, with the rest of the paper detailing design choices for said features, implementation details, and practical applications for some simple student projects.